

Title: **Evaluation And Attribution Of OCO-2 XCO₂ Uncertainties**
Authors: John Worden[1], Gary Doran[1], Susan Kulawik[2], Annmarie Eldering[1], David
Crisp[1], Christian Frankenberg[3,1], Chris O'Dell[4], and Kevin Bowman[1]
1)  Jet Propulsion Laboratory / California Institute for Technology
2)  BAERI research
3)  California Institute for Technology
4)  Colorado State University
**Abstract**
Evaluating and attributing uncertainties in total column atmospheric $CO_2$
measurements ($XCO_2$) from the OCO-2 instrument is critical for testing hypotheses
related to the underlying processes controlling $XCO_2$ and for developing quality flags
needed to choose those measurements that are usable for carbon cycle science.
Here we test the reported uncertainties of Version 7 OCO-2 $XCO_2$ measurements by
examining variations of the $XCO_2$ measurements and their calculated uncertainties
within small regions (~100 km x 10.5 km) in which $CO_2$ variability is expected to be
small relative to variations imparted by noise or interferences.  Over 39,000 of
these "small neighborhoods" comprised of approximately 190 observations per
neighborhood are used for this analysis. We find that a typical ocean measurement
should have a precision and accuracy of 0.35 and 0.24 ppm respectively for
calculated precisions larger than ~0.25 ppm. These values are approximately
consistent with the calculated errors of 0.33 and 0.14 ppm for the noise and
interference error (assuming that the accuracy is bounded by the calculated
interference error). The actual precision for ocean data becomes worse as the
signal-to-noise increases or the calculated precision decreases below 0.25 ppm for
reasons that not well understood.  A typical land measurement (both nadir and
glint) is found to have a precision and accuracy of approximately 0.75 ppm and 0.65
ppm respectively as compared to the calculated precision and accuracy of



approximately 0.36 ppm and 0.2 ppm. However, this precision includes the effects of
synoptic variability in the total column that could be as high as 0.5 ppm during the
summer drawdown period. The accuracy is likely related to interferences such as
aerosols or surface albedo and is a lower bound as it is evaluated by comparing
gradients in OCO-2 estimates of $XCO_2$ to expected gradients across the region and
not by direct comparison to well-calibrated $XCO_2$ measurements from the ground
network.
**1.0 Introduction**
Variations of total column $CO_2$ ($XCO_2$) resulting from photosynthesis and
respiration in tropical forests (e.g. Parazoo et al. 2013), urban emissions (e.g. Kort et
al., 2012) or tropical fires (e.g. Bloom et al., 2016) range from 2 – 5 ppm.
Consequently, in order to use space-based measurements of $XCO_2$ to infer fluxes or
properties of the processes controlling these variations, uncertainties in $XCO_2$
should ideally be much much smaller than this variability (Miller *et al.* 2007). The
Orbiting Carbon Observatory-2 (OCO-2) was launched in July 2014, to measure the
atmospheric column averaged carbon dioxide ($CO_2$) dry air mole fraction, $XCO_2$ with
the precision, accuracy, and coverage needed to quantify variations on regional
scales at monthly intervals.  These measurements are being used to investigate the
underlying carbon cycle processes controlling atmospheric $CO_2$.  The radiative
transfer and $XCO_2$ estimation (or retrieval) algorithms (Boesch et al. 2006; 2011;
Connor et al. 2008; O'Dell et al., 2012) were developed and tested using observed
radiances from the Japanese TANSO GOSAT instrument (Kuze et al. 2009; Yoshida et
al. 2011), which measured similar spectral regions as the OCO-2 mission.  These
algorithms also allowed extensive evaluation of quality flags and metrics needed to
reject estimated $XCO_2$ values which were clearly spurious, likely because of poorly
estimate values for aerosols, clouds, surface albedo or surface pressure (Crisp et al.,
2012; Mandrake et al., 2013).  In this paper we evaluate the calculated uncertainties
due to noise and interferences in the OCO-2 data product (Version 7).



Our approach follows the methodology described in Boxe et al. [2010] and Kuai
*et al.* [2013] in which variations of the observed trace gas over a small "area" are
compared to the calculated errors. Figure 1 shows the distribution of latitudinal
gradients in $XCO_2$ over the ocean and over North America based on the "high
resolution" Carbon Tracker model (e.g. Peters *et al.*, 2007) with ~100 km spatial
resolution. This distribution is calculated by differencing $XCO_2$ from adjacent model
grid points, as a function of latitude, using all modeled $XCO_2$ values in July 2015. We
find that the root-mean-square (RMS) value of these gradients is approximately 0.3
ppm/100 km during the summer and ~0.1 ppm/100 km during November. Keppel-
Aleks [2011, 2012] also found North American summertime gradients in *$XCO_2$*
between 0.1 ppm/100 km to 0.3 ppm/100 km using ground based total column data
and measured wind speeds. In addition, these studies found synoptic variability
could change $XCO_2$ values by up to 0.5 ppm over the study time period in a random
manner (Figure 5 in Keppel-Aleks [2011]). In contrast, Figures 1a and 1b show that
typical variations in the gradients over the ocean should be less than that of land,
between ~0.1 ppm/100 km to 0.2 ppm/100 km. While in situ measurements [e.g.
Wofsy *et al.*, 2011] and model data do show variations in $XCO_2$ that are sometimes
larger than 0.2 ppm/100 km we would expect that these variations do not represent
typical *$XCO_2$* gradients, especially since the total column of $CO_2$ integrates the effects
of many sources and sinks from hundreds to thousands of kilometers away from the
observation [e.g. Keppel-Aleks *et al.*, 2011]. Because the expected variability in
$XCO_2$ from models, ground-based data, and in situ measurements are comparable or
less than the calculated OCO-2 uncertainties, we can compare the observed
variability of $XCO_2$ from OCO-2 data within a small region, covering an orbit track
that spans 100 km in latitude, to evaluate the magnitude and character of their
corresponding calculated uncertainties.
**2.0 Overview of OCO-2 data**



The OCO-2 instrument measures radiances in the molecular oxygen ($O_2$) A-band
(0.765 microns), the "weak" $CO_2$ band at 1.61 microns and the "strong" $CO_2$ band at
2.06 microns.  The OCO-2 instrument is an imaging spectrometer that collects with 8
samples, or "spatial footprints" across a narrow (0.8-degree) swath track observes
near the "glint spot" where sunlight is specularly reflected by the surface.
Observations are taken in three different modes, (1) "Nadir", where the space-craft
points the instrument's aperture at the ground directly downward along the orbit
track, (2) "Glint," where the space craft points instrument's aperture near the "glint
spot" where sunlight is specularly reflected by the surface, near the specular
reflection point for sunlight, and (3) Target, where the space-craft points the
instrument aperture at a stationary surface target, such as a validation site or city.
Nadir observations usually return useful measurements only over land.  Glint
observations return useful data over both land and ocean. Here, we discriminate
land-glint and ocean glint observations because they have different error statistics.
We do not evaluate Target data in this analysis due to spurious statistics that are
observed with the Target data.
As discussed in Boesch et al. [2006]; Connor *et al.*  [2008] and O'Dell *et al.*  [2012
and references therein] total column estimates of *XCO₂*, are derived from OCO-2
observed radiances using a Bayesian optimal estimation approach that depends on
$CO_2$, all the geophysical parameters or interferences that affect the radiances in
these bands, and *a priori* statistics of the atmosphere and these interferences.
We use version 7 of the OCO-2 data, the first OCO-2 product distributed for
general users. These data, like those described  for GOSAT data in Wunch *et al.*
[2011], are bias corrected based on comparisons between OCO-2 and total column
measurements from the ground-based Total Carbon Column Observing Network
(TCCON). Data quality is evaluated using a variety of metrics that depend on the
estimated cloud, aerosol, and surface properties, convergence and known statistics
of the retrieved $CO_2$ values (e.g. Mandrake *et al.* , 2013). Data quality flags are given
as "warn levels" with values ranging from 0 (best) to 20 (worst).  Data with lower
warn levels are more likely to represent the statistics of the observed $CO_2$ whereas





data with higher warn levels likely or are too strongly affected by interfering effects.
The warn levels are primarily evaluated empirically; for these reasons we
conservatively use only data with warn levels of 10 or smaller to ensure that the
corresponding errors are likely well characterized:
http://disc.sci.gsfc.nasa.gov/OCO-2/documentation/oco-2-
v7/OCO2_XCO2_Lite_Files_and_Bias_Correction_0915_sm.pdf.
**3.0 Evaluation of Uncertainties**
*3.1 Overview of Error Analysis and Methodology*
We evaluate the uncertainties of the *XCO2* observations by examining the
variations of $_{XCO2}$ within small neighborhoods of approximately 10.5 km by 100 km
in size. After warn level filtering, this "small neighborhood" test set is composed of
approximately 1.5 million Land-Nadir soundings, 1.0 million Land-Glint soundings,
and 5.0 million Ocean-Glint soundings. Each neighborhood contains at least 50
soundings, with roughly 190 soundings per neighborhood on average, and
approximately 39,000 small neighborhoods in total across the three modes.
stretching from approximately 30S to 30N. The strict filtering used in this analysis
(Warn Levels <= 10), and the need for at least 50 measurements per bin limits this
analysis to latitudes between 30S to 30N, primarily over drier, sub-tropical regions
over land but no obvious preferential distribution over the ocean (not shown).
As discussed in [O'Dell et al., 2012], a $CO_2$ profile is simultaneously estimated
with all other geophysical parameters that affect the observed radiance such as
aerosols, albedo, and surface pressure.  The "column-averaged dry air mole fraction"
of $CO_2$ or $XCO_2$ is then calculated by applying the column operator [e.g. Connor et al.,
2008; Worden et al., 2015] to the estimated $CO_2$ profile. As discussed in Rodgers
[2000], Worden *et al.*  [2004], Connor [2008], and Bowman *et al.* [2006], when this
non-linear retrieval converges to a solution, the estimated $XCO_2$ can be written as:

$$\hat{X} = X_a + h^T A_{xx}(x - x_a) + h^T A_{xy}(y - y_a) + h^T G n + h^T G \sum_i K_i \delta_i \quad (1)$$
where $\hat{X}$ is the estimated total column for $CO_2$, $\hat{X}_a$ is the *a priori* value used to help
regularize the retrieval, the vector $x$ is the "true" $CO_2$ profile in units of volume
mixing ratio (VMR), discretized onto the forward model atmospheric pressure grid
used to calculate the transfer of radiation needed to model the observed radiance.
The $x_a$ is the *a priori* for the $CO_2$ profile. The vector "$y$" contains all the other
parameters that are simultaneously estimated with $x$ such as aerosol properties,
surface albedo, surface pressure. The vector "$n$" is the actual noise in the radiance.
The quantities $x, y,$ and $n$ are not known exactly, only their statistical properties can
be estimated. The vector "$h$" is the column operator which maps a profile on the
pressure grid defined by "$x$" into a dry air total column. The averaging kernel matrix
$A$ describes the sensitivity of the estimate to each retrieved parameter [Rodgers,
2000]. In equation 1 the averaging kernel matrix is composed of two parts, $A_{xx}$ and
$A_{xy}$, described by:
$$A = \begin{bmatrix} A_{xx} & A_{xy} \\ A_{yx} & A_{yy} \end{bmatrix} \qquad\qquad (2)$$
For example $A_{xx}$ describes the sensitivity (or $\frac{\partial \hat{x}}{\partial x}$) of the estimated $CO_2$ on each level,
$x$, to its true value, whereas $A_{xy}$ describes the sensitivity of the estimated $CO_2$ on
each level, $x$, to all other simultaneously estimated parameters, e.g., aerosols, etc.
The matrix, "$G$," is the gain matrix, , which is the derivative of the estimated $CO_2$ on
each level, x, to the observed radiance, "$L$" (or $G = \frac{\partial \hat{x}}{\partial L}$). The matrix, "$K$," is the
Jacobian, or sensitivity of the observed radiance to a parameter (e.g. $K = \frac{\partial L}{\partial x}$). The last
term, $\delta$, describes the error in all parameters that are not estimated for this
retrieval, but are assumed constant, such as absorption coefficients or instrument
functions (e.g. Connor et al., 2008). The mean $CO_2$ column is written as:



$\hat{X}_{mean} = X_a + h^T \frac{1}{N} \sum_{j=1}^{N} A_j (x_j - x_a) + \frac{1}{N} h^T \sum_{j=1}^{N} A_{xy}^j (y_j - y_a) +$

2       $\frac{1}{N} \sum_{j=1}^{N} h^T G_j (n_j + \sum_{i,j} K_{i,j} \delta_{i,j})$                                                  (3)

where N is the number of observations within the small area and for simplicity we
assume the column operator $h$ is constant across the domain.
For the next three sections, we test the following hypotheses regarding the observed
distributions within the collection of "small neighborhoods" and their calculated
uncertainties:
H1: Uncertainties within a small area are primarily due to random noise
H2: Uncertainties are correlated
H3: Uncertainties within a small area are described by a slowly varying bias
(consistent with the expected effects of interference error).
We look at the variability with respect to the neighborhood mean in two ways: (1)
for small neighborhoods; the predicted errors for a neighborhood are averaged
from the observations that comprise that neighborhood, making the statistics
technically a sum of Gaussians, and (2) the variability with respect to the
neighborhood mean, sorted by predicted error and aggregated over many
neighborhoods; the statistics in this case should be Gaussian, however the locality of
the analysis is somewhat reduced.
*3.2 H1: Error due to noise*

26       To evaluate whether measurement noise in the radiances is the primary

factor driving variability within a small area we assume that the terms $A_{xy}(y_j\text{-}y_a)$
and systematic errors $K_{i,j}\delta_{i,j}$ do not vary. Based upon these approximations, the
difference between an observation and its mean is given by:





$\quad \widehat{X}_{obs} - \widehat{X}_{mean} = \ \delta_{obs} = \delta_{XCO2} + \mathbf{G_{obs}}\boldsymbol{n_{obs}} - \frac{1}{N}\sum_{j}^{N}\mathbf{G_j}\boldsymbol{n_j}$ (4)
where $\delta_{XCO2} = \boldsymbol{h}^T\mathbf{A}(x_{obs} - x_{mean})$ and is the difference between the individual
"true" $XCO_2$ and the mean of the "true" $XCO_2$ values within the neighborhood.
Assuming the measurement noise is spatially uncorrelated, the variance within the
small neighborhood is [e.g. Bowman *et al.*, 2006] is:
$\quad \boldsymbol{Var}\left\|\widehat{X}_{obs} - \widehat{X}_{mean}\right\| = \sigma_{obs}^2 = \sigma_{XCO2}^2 + \ \sigma_{noise}^2 + \frac{1}{N^2}\sum_{j=1}^{N}\sigma_j^2 - \frac{2}{N}\sigma_k^2$ (5)
where $\boldsymbol{\sigma_{noise}} = \mathbf{G_K}\mathbf{S_k}\mathbf{G_K^T}$ is the measurement uncertainty due to noise. The $\sigma_{XCO2}$ is
the variability of the true $XCO_2$ within the small neighborhood. The $\mathbf{S_k}$ is the
spectral instrumental noise covariance and is calculated during calibration of the
instrument. The individual $\sigma_{noise}$ values are provided for each measurement in the
OCO-2 product files. For large N, Equation 5 is approximately equal to:
$\quad \sigma_{XCO2}^2 + \ \sigma_{noise}^2.$
We next evaluate these uncertainties using two approaches. In the first
approach we gather all observations that have approximately the same calculated
measurements uncertainty, $\sigma_{noise}$, (to within 0.01 ppm) as provided in the OCO-2
product files and compare to the actual variability of these observations. The steps
for this comparison are:
1) Calculate the $\delta_{obs}$ or difference between an observation and its mean
within a small neighborhood as shown in Equation 4.
2) Collect all of the $\delta_{obs}$ values from all neighborhoods used in this analysis
whose corresponding $\sigma_{noise}$ values (measurement uncertainty) are the
same to within 0.01 ppm and bin them as a function of $\sigma_{noise}$. There are
typically about 1000 observations per $\sigma_{noise}$ bin.

1    3) Compare the standard deviation of the collection of $\delta_{obs}$ values within

2        each bin to the expected standard deviation due to noise or, $\sigma_{noise}$. Based

3        on Equation 5 we should expect to get a linear, one-to-one relationship if

4        the dominant parameter affecting the variability within a small

5        neighborhood is noise.

The results of these comparisons for land-nadir, land-glint, and ocean-glint
observations are shown in the upper left panels of Figures 2, 3, and 4 respectively.
These results show the calculated measurement error has skill, i.e. there is a linear
relationship between calculated and actual error. However, over land the observed
random variability is approximately 0.4 ppm larger than the variability expected
from noise. Synoptic variations in $XCO_2$ could potentially explain much of this extra
0.4 ppm however other sources of variability could be due to the strong non-
linearities in the retrieval [e.g. Kulawik *et al.*, 2008] or local variability between the
true and *a priori* in the interferences, or non-retrieved parameters. Over the ocean
there appears to be an even stronger one-to-one relationship between the
calculated uncertainty and the actual uncertainty except for calculated uncertainties
less than approximately 0.25 ppm which show a strong inverse relationship. We
find that these observations (not shown) tend to occur in the tropics in cloudy
regions and that the observations tend to have very high signal-to-noise ratios.

21       We next test whether the calculated measurement noise is a useful value for

predicting the expected distribution of observations within a neighborhood.
Because each $\delta_{obs}$ is drawn from a distribution with a different variance, we treat
the sample of each set of observations, $[\delta_1, \delta_2, \dots \delta_N]$, as being drawn from an
uncorrelated distribution with individual variances $\sigma_{obs}^2$. Accordingly, the variance
of this sample should be the average of the individual variances $\sigma_{obs}^2$:
$$\boldsymbol{Var}|[\widehat{\boldsymbol{X}}_{obs} - \widehat{\boldsymbol{X}}_{mean}]|| = \left\|[\delta_1, \delta_2, \dots \delta_N]\right\| = \frac{1}{N}\sum_j^N \boldsymbol{\sigma}_j^2 \qquad (6)$$





The top right panel of Figure 1 shows a comparison of the observed variance of the
$XCO_2$ distributions  (using the left side of Equation 6) within each neighborhood
(black circles) versus the expected variance  in $XCO_2$ using the right side of Equation
6. Each black symbol represents a single neighborhood. In contrast to the top left
panel of Figure 1, this result suggests that the measurement error has no skill in
predicting the observed variance of $XCO_2$ within a neighborhood.
We next test whether the observed variance, versus that due to measurement
noise or sampling, explains the upper right panel of Figures 2, 3, and 4.  To perform
this test, we perform the following steps:
1)  Within each neighborhood, replace the calculated measurement error with

12        the "actual" measurement error as shown by the solid red line in the upper

13        left panel of Figures 2, 3, and 4, for each observation.

2)  Create a simulated distribution of observations based on this new

15        uncertainty.

3)  Randomly sample (or take) one of these observations → label this the

17        "modeled" observation.

4)  Repeat steps 1-3 for all observations in the neighborhood.
5)  Calculate the variance of this "modeled" set of observations for each

20        neighborhood.

The red dots in Figures 2b, 3b, and 4b show the modeled distributions using the
steps discussed above. The modeled distribution is more consistent with the mean
of the observed distribution relative to the one-to-one line. However, it is clear from
this simulation that errors due to random noise and sampling do not explain the
observed variance for each neighborhood although the distribution of variances for
the ocean show much better agreement relative to the land distributions.
*3.3 H2: Uncertainties are correlated*





We next test whether observed correlations in the data could explain the
distributions of the data within a neighborhood. Figures 5 shows the joint
distribution of the $XCO_2$ anomaly and a 0.3 second lagged anomaly in a
neighborhood. If the data were uncorrelated then the joint distribution should be
circular; the asymmetric distribution therefore implies that the errors, as
empirically described by the differences, are correlated.  Figures 6a and 6b show
that autocorrelation is observed both in time for measurements made on the order
of 1 second of each other, and with respect to the spatially adjacent "footprints," the
8 simultaneous measurements made by the OCO-2 instrument at each time.  The
range of correlations for the different observation types, land nadir, land glint, and
ocean glint are 0.45, 0.43, and 0.28 as a function of footprint and 0.31, 0.34, and 0.24
as a function of time.

13        In order to test whether these observed correlations could explain the

distributions shown in Figures 2, 3, and 4, we conservatively use a correlation
coefficient of 0.7 for all observations (an extreme case). We then use the following
procedure, building on the steps described in the previous section.
1) Within each neighborhood replace the calculated measurement error with

18         the "actual" measurement error as shown in the upper left panels of Figures

19         2, 3, and 4 for an observation

2) Starting with the first observation (in time) within a neighborhood for

21         Footprint #1, sample a value for the observation from the distribution of

22         "actual" measurement errors. Label this the "modeled" observation.

3) For all subsequent observations in time for Footprint #1, sample each

24         "modeled" observation from a distribution that is correlated with the

25         modeled observation at the previous time step and has a variance

26         corresponding to the "actual" measurement error.

4) For observations in Footprints #2-8, sampling each modeled observation

28         from a distribution correlated with the modeled observation at the same

29         time step in the previous (adjacent) footprint, again with a variance

30         corresponding to the "actual" error.





5) Calculate variance of this "modeled" set of observations, for each

2        neighborhood.

As can be seen in the lower left panels of Figures 2, 3, and 4, adding correlations to
the data makes the comparison worse because the modeled distributions become
much narrower relative to the modeled distributions in the upper right panels of
these figures.  Our conservative choice of a 0.7 correlation between observations at
adjacent times and footprints illustrates this effect clearly. We therefore conclude
that while correlations are empirically observed in the data, they cannot completely
explain the observed distributions within the small neighborhoods.
*3.4 H3: Uncertainties within a small area are characterized as a slowly varying bias.*

14       We next examine whether "non-random" uncertainties could explain the

observed distributions in the upper right panels of Figures 2,3, and 4.  For example,
as shown in Equation (1), the jointly retrieved parameters ($y - y_a$) might remain
constant across a neighborhood but the Averaging kernel associated with this term,
which is given by $\mathbf{A_{xy}} = \frac{\partial \mathbf{x}}{\partial \mathbf{L}} \frac{\partial \mathbf{L}}{\partial \mathbf{y}} = \mathbf{GK_y}$, can vary across a neighborhood as the pointing
angle varies. The effect of non-retrieved parameters such as instrument effects or
spectroscopy on the estimate can vary for the same reason.

21       Figure 7 shows the variation of $XCO_2$ across one of the ocean neighborhoods

for all 8 OCO-2 footprints (denoted by "FP"). The right panel shows the observed
distribution in black relative to the mean $XCO_2$ of the neighborhood. For reference,
the red dashed line in the right panel indicates the expected distribution if only
random noise explained the variability. The slope shown in Figure 7 represents an
extreme case but demonstrates that observations can pass the set of quality flags
but still show this unlikely behavior over the ocean.  Figure 8 shows the distribution
of all slopes across all land-nadir neighborhoods used in this study and different fits
(Gaussian, Lorentz, Laplace) to the distribution. The Laplace distribution provides



the best overall fit so we use its functional form as a simple, convenient description of the shape of the sharply peaked slope distribution. More complex models such as Gaussian mixtures might also describe the shape of this distribution of slopes as drawn from several distinct "populations" of neighborhoods, but we leave such an analysis to future work. The RMS of the distribution is approximately 1.28. which is much larger than expected variations in $XCO_2$ (e.g. Figure 1 and Keppel-Aleks *et al.* [2012]).

For land-nadir, land-glint, and ocean-glint data the variance of the slopes is given by 1.28 ppm/100 km, 1.12 ppm / 100 km, and 0.48 / 100km respectively. To test whether these slowly varying changes explains the distribution of $XCO_2$ within small neighborhoods we follow the same steps described in Section 3.2 and 3.3 but now add another:

1) Within each neighborhood replace the calculated measurement error with the "actual" measurement error as shown in the upper left panels of Figures 2, 3, and 4 for an observation

2) Starting with the first observation (in time) within a neighborhood for Footprint #1, sample a value for the observation from the distribution of "actual" measurement errors. Label this the "modeled" observation.

3) For all subsequent observations in time for Footprint #1, sample each "modeled" observation from a distribution that is correlated with the modeled observation at the previous time step and has a variance corresponding to the "actual" measurement error.

4) For observations in Footprints #2-8, sampling each modeled observation from a distribution correlated with the modeled observation at the same time step in the previous (adjacent) footprint, again with a variance corresponding to the "actual" error.

5) Adjust each modeled observation with a linear function where the slope of the linear function is randomly chosen from the fitted Laplace distribution to the slopes (e.g., the Laplace function shown in Figure 8)





6) Calculate variance of this "modeled" set of observations, for each

2       neighborhood.

Figures 2, 3, and 4 (lower right panels) show the best overall agreement
between modeled distributions of $XCO_2$ relative to the mean and the expected
distributions based on observations, demonstrating that a slowly varying bias is
needed to best explain the observed distributions within a grid of approximately
100 km x 10 km.

9       The expected "true" variability across a typical 100 km neighborhood is ~0.1 to

~0.3 ppm (e.g. Figure 1). Each typical observation has a random error related to
noise and a systematic error that is in principal bounded by the calculated
interference error (e.g. Boxe *et al.*, 2010) and is approximately 0.2 ppm. The 100 km
x 10.5 sizes for the small neighborhoods used for this analysis is a fortuitous size
because the expected latitudinal variability is approximately the same or smaller as
the mean interference error (Figure 1). Within a typical grid box an OCO-2 observed
measurement over land is within 1.28 / 2, or ~0.65 ppm of the mean $XCO_2$ value.
For these reasons, and we expect that a typical observation over land has at least a
systematic error of at least 0.65 ppm, about 2 to 3 times larger than the calculated
interference error.
In contrast, the observed distributions of slopes and (mean slope of 0.48 ppm /
100 km or mean error of 0.24 ppm) for the ocean data is only 70% larger than the
mean calculated interference error of 0.14 ppm.  Because the distribution of ocean
data within "bins" (Figure 4, upper left panel) is also well described by the
calculated random error, we conclude that the ocean glint data is reasonably well
characterized by its calculated uncertainties for this size of a grid box, except for
calculated noise (or precision) uncertainties that are less than ~0.25 ppm.

27       We find no relationship between the distribution of slopes for a neighborhood

and the corresponding mean of the calculated interference error suggesting that the
calculated interference error does not explain the observed slope within a
neighborhood, in contrast to the measurement error.  However, there is a





correlation between the slope and the estimated magnitude of interferences, such as
aerosol optical depth, surface albedo, and surface pressure.  For example, the
correlation between the slopes of land-glint data with the mean uncertainty in the
interferences is 0.06 whereas the correlation between the observed slopes in $XCO_2$
and similarly calculated observed slopes in aerosol optical depth is 0.37. This
correlation suggests that the observed slow variations in $XCO_2$ across a
neighborhood could be related to how interferences affect the $XCO_2$ estimate as
OCO-2 takes observations across a neighborhood.
4.0 **Summary**
The analysis described in this paper uses the observed $XCO_2$ variability across
small neighborhoods, in comparison to expected variations, to evaluate the
precision and accuracy of the $XCO_2$ data. We find that the precision and accuracy of a
typical ocean measurement is approximately 0.35 and 0.2 ppm respectively,
consistent with the calculated errors (assuming that the accuracy is bounded by the
calculated interference error and does not include smoothing error).  The precision
and accuracy of a typical land measurement (both nadir and glint) is approximately
0.75 ppm and 0.65 ppm.  These values can be compared to the calculated
measurement and interference errors of approximately 0.36 ppm and 0.2 ppm.
Much of the difference between the observed precision and calculated measurement
error could be due to natural synoptic variability in $XCO_2$ but is also likely due to
non-linearities in the retrieval or random components of interference error. The
accuracy is estimated from observed gradients in $XCO_2$ of approximately 1.28 ppm /
100 km across the small neighborhoods used in this analysis. Natural variability can
likely explain at most about 0.1 to 0.3 ppm of this of 1.28 ppm. The accuracy is
estimated as being at least half the value of this slope or ~0.65 ppm.
This 0.65 ppm estimate for the accuracy of the land data could be a lower bound
because it is based on observed gradients across a region and not direct
comparisons against TCCON, although the OCO-2 data are bias corrected using



TCCON data (Wunch *et al.* 2011). We find a relationship between these gradients
and interferences such as aerosol optical depth and surface albedo suggesting that
these interferences are the cause of the gradients.
The analysis discussed in this paper can be applied to future versions of the
OCO-2 data in which more accurate calculations of the interferences are included or
additional data quality flags are used to remove spurious individual observations or
sets of observations. For example, another set of data quality flags could be
developed to remove observations that vary too much over a region. In addition,
Connor *et al.* (2016, submitted) finds that other instrumental and spectroscopic
uncertainties need to be included in the error analysis and that these additional
components will likely have a random and systematic component, thus possibly
explaining the discrepancy between calculated and actual uncertainties discussed
here. A future study in which the calculated uncertainties discussed in Connor *et al.*
(submitted) repeats the steps shown in this paper could be of great value for
explaining the observed variations across the small neighborhoods used in our
analysis.
**Acknowledgements**
Part of this research was carried out at the Jet Propulsion Laboratory, California Institute
of Technology, under a contract with the National Aeronautics and Space Administration.
Funding for Susan Kulawik provided by NASA Roses NMO710771/NNN13D771T,
"Assessing OCO-2 predicted sensitivity and errors". CarbonTracker NT-NRT results
provided by NOAA ESRL, Boulder, Colorado, USA from the website at
http://carbontracker.noaa.gov.

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






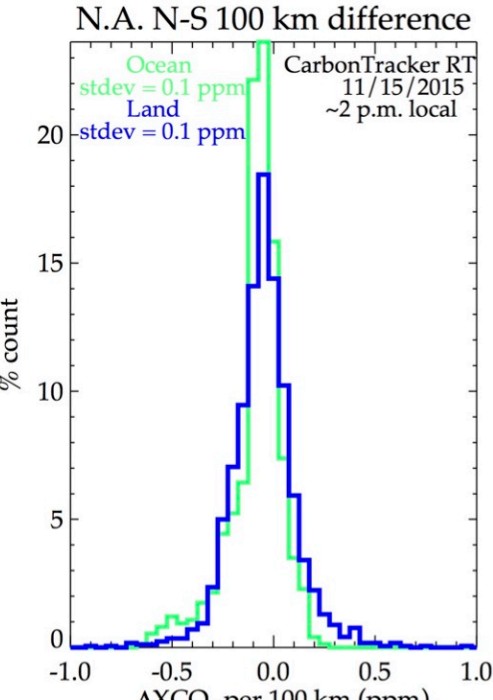 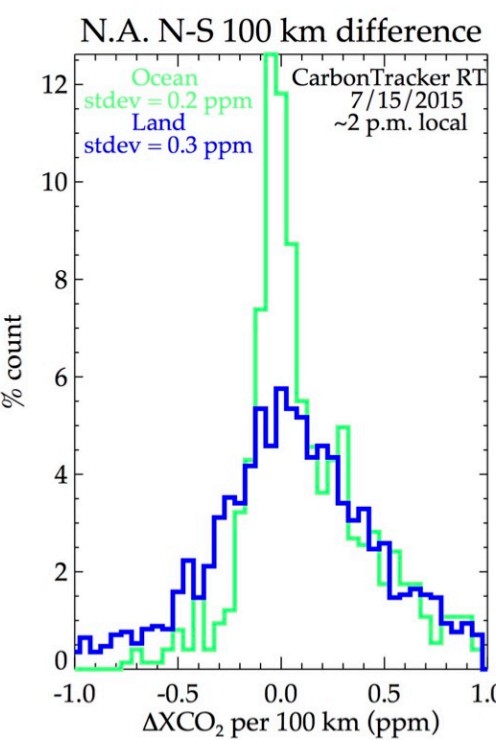

Figure 1: Distribution of latitudinal XCO$_2$ gradients as calculated by the high resolution, "Real
Time", Carbon Tracker model for November 2015 (left panel) and July 2015 (right panel) over
North America and the nearby oceans. The latitude grid is 1 degree or ~110 km. The gradients
are re-scaled to 100 km for comparison to the XCO$_2$ gradients discussed in this paper.





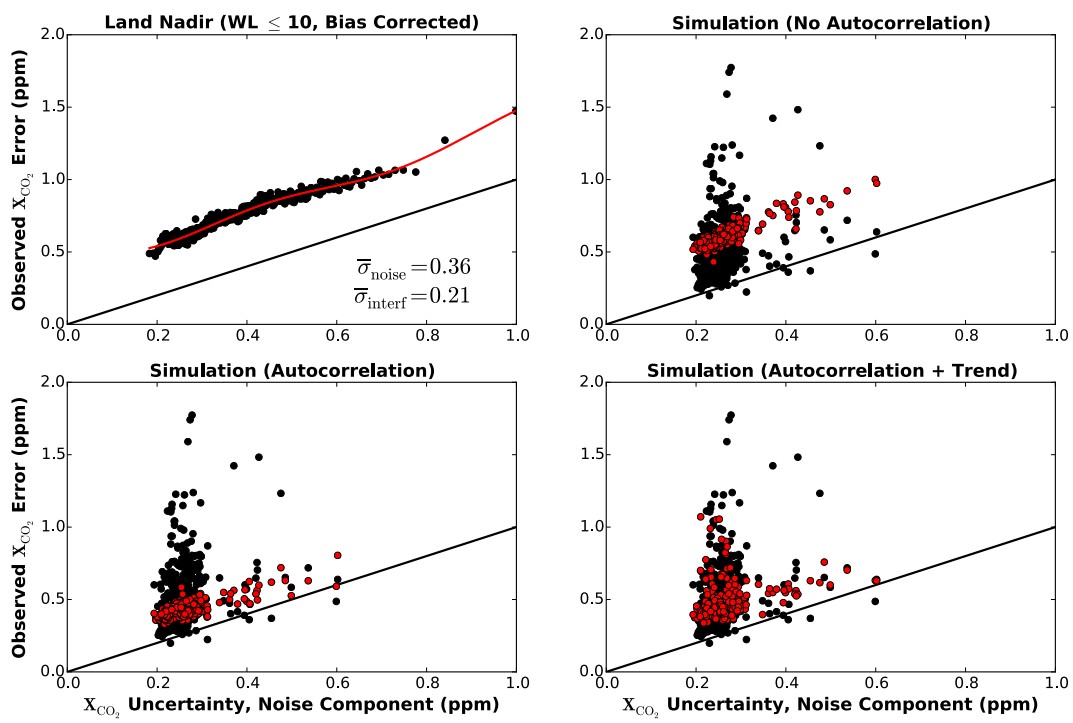

Figure 2: Calculated, observed, and modeled uncertainties for Land-Nadir observations. Black
circles are the observed distributions and red circles are modeled distributions assuming
sampling and random error (upper right), correlated errors (bottom left) and correlated plus
trend in error (bottom right).




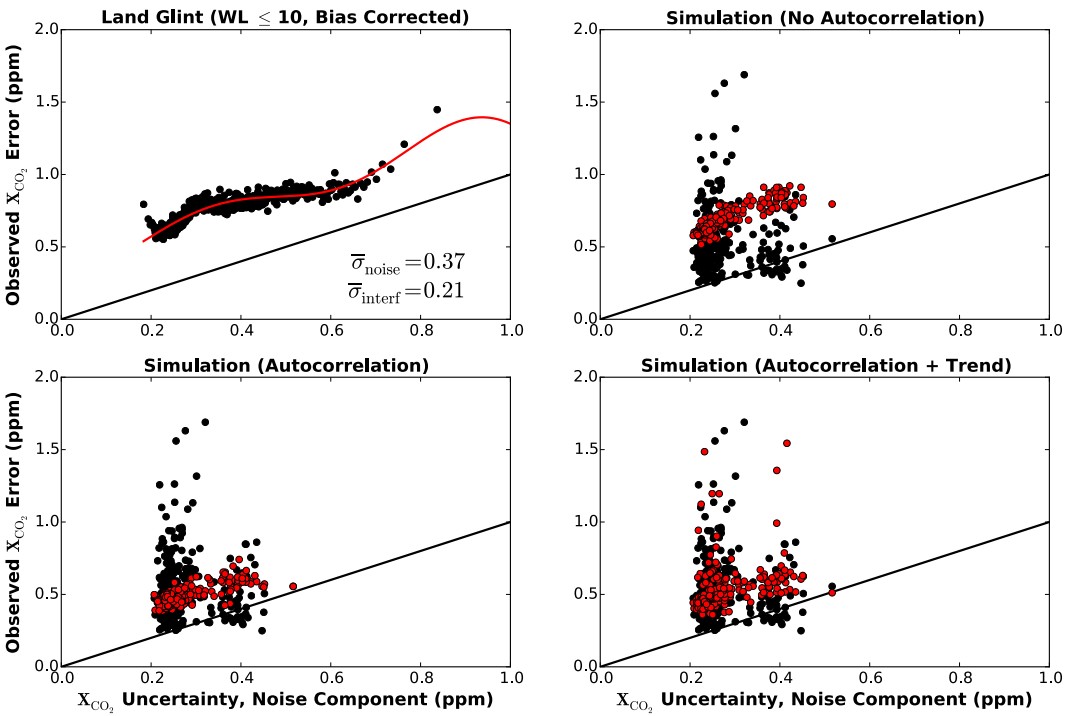

2
3    Figure 3: Observed and modeled distributions for Land-Glint data





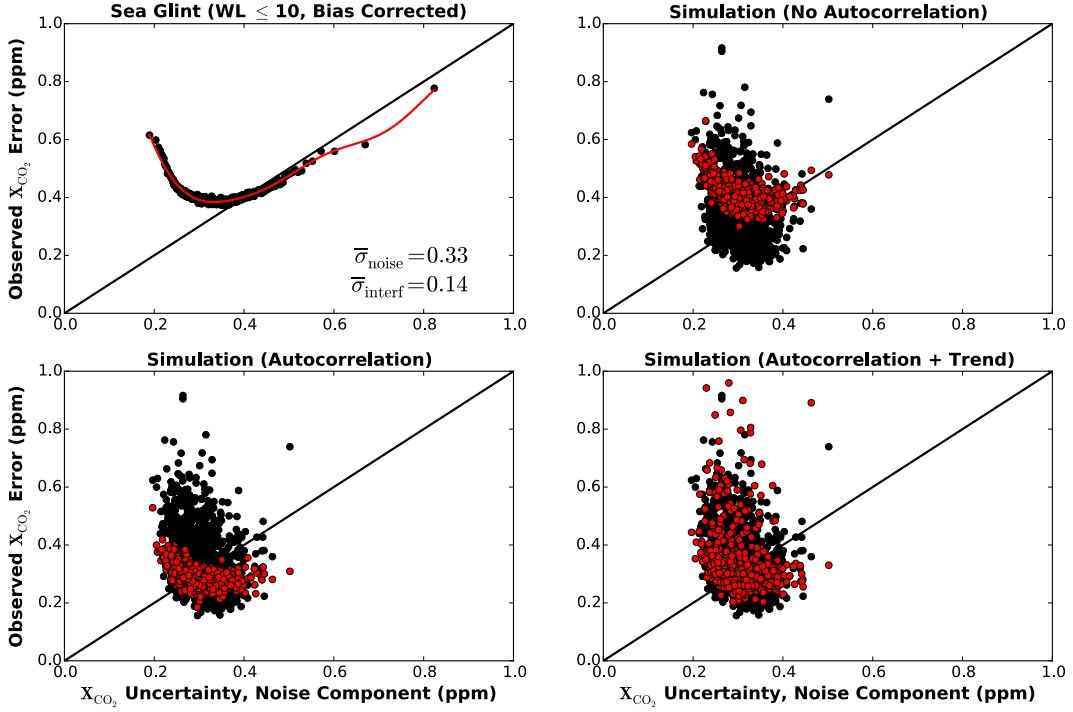

2
3    Figure 4: Observed and modeled distributions for Sea-Glint data.
4




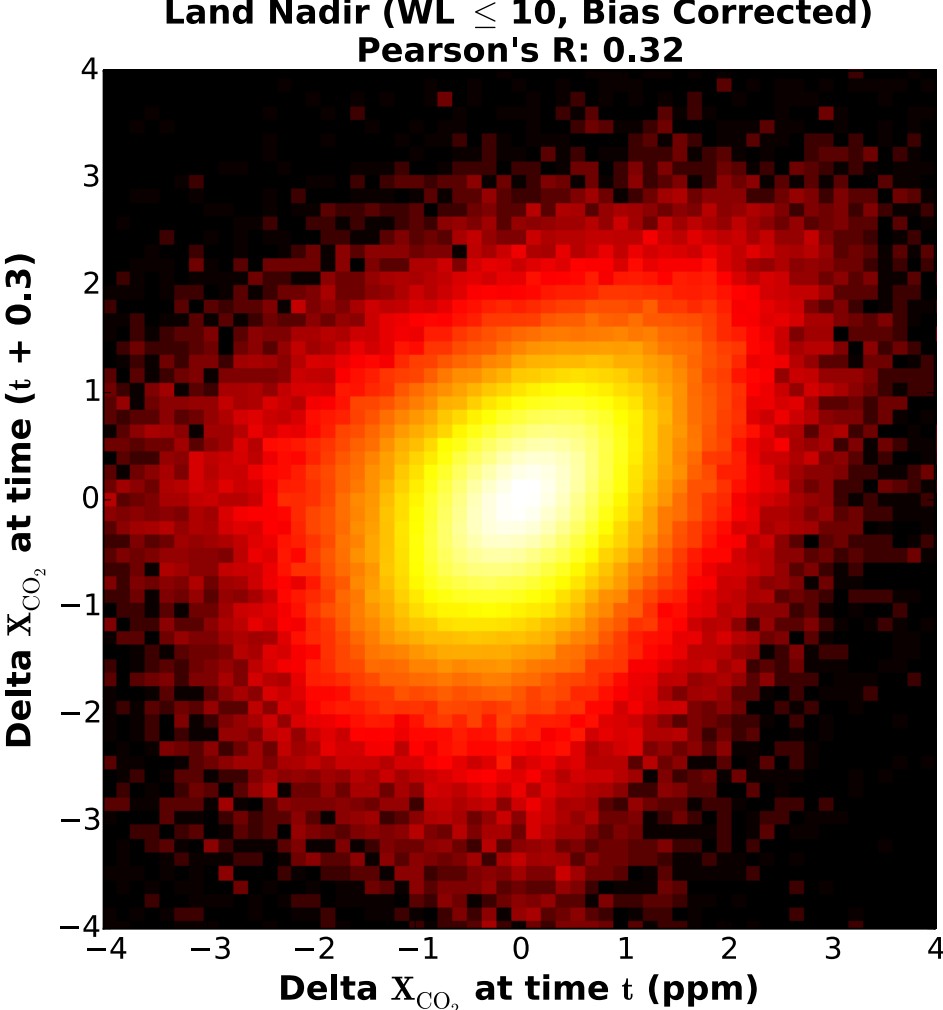

Figure 5: Distribution of XCO$_2$ values between time steps for the set of observations from each
"small neighborhood" used in this analysis.




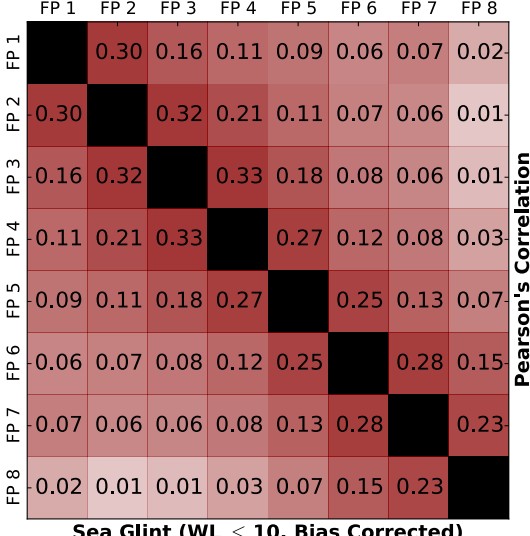

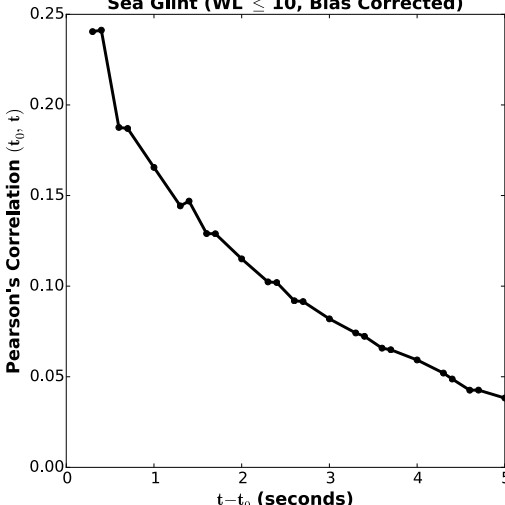

Figure 6: (Top) Correlation of differences across pixels between observed minus mean within a
neighborhood. (Bottom) correlation between observations for a single pixel.



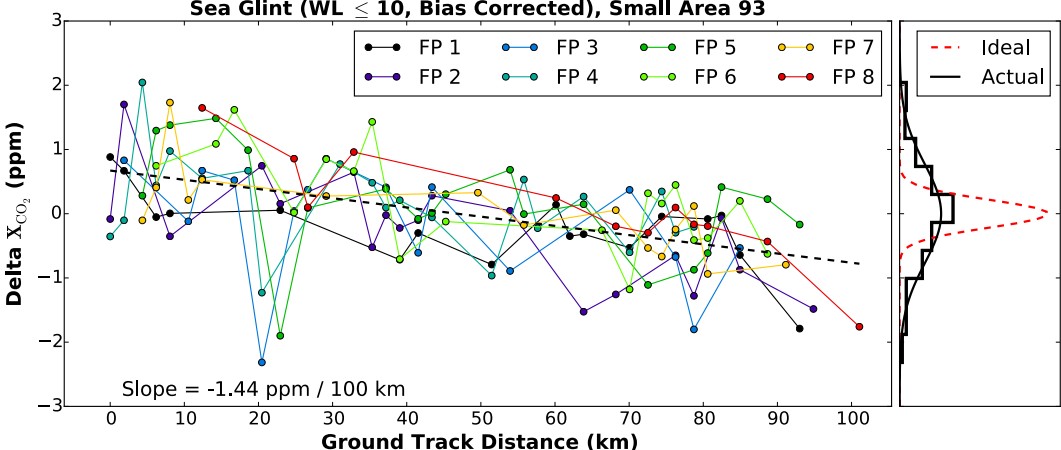

Figure 7:  The difference between XCO₂ and the mean value for one of the small neighborhoods
(or areas) used in this analysis. The left panel shows the differences for each footprint (FP),
representative of one of the OCO-2 observations. The right panel shows the observed
distribution (actual) and one calculated if the distributions were representative of the
calculated random error.





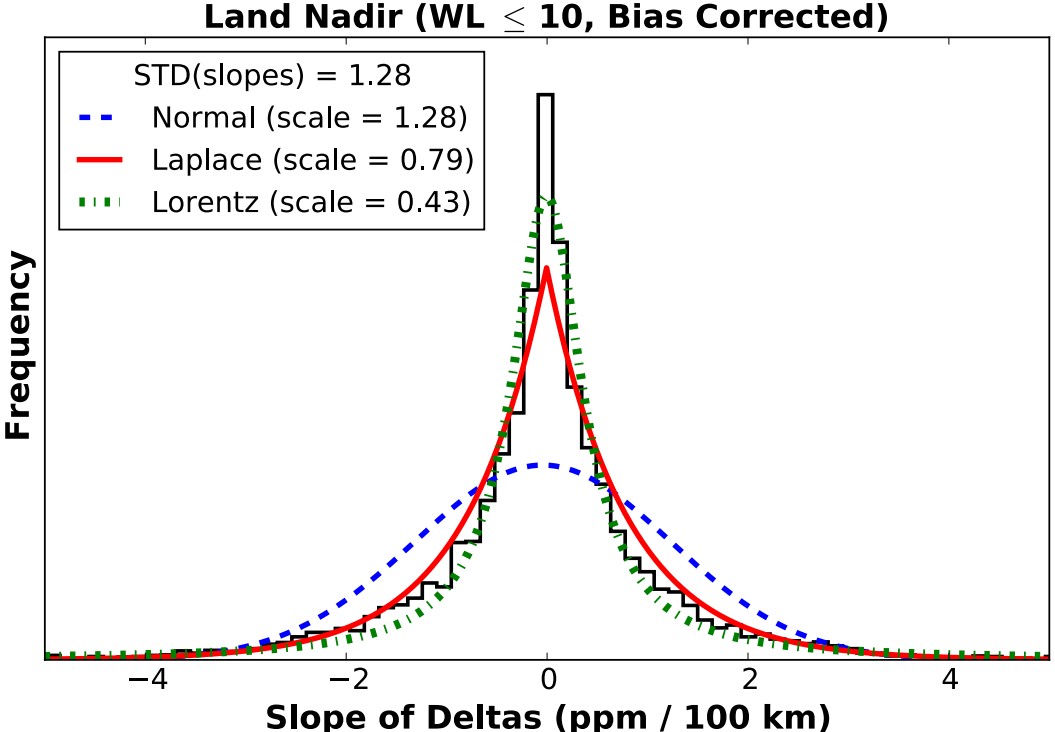

3  Figure 8: The distributions of slopes of the observed XCO$_2$ gradients across all the small
4  neighborhoods corresponding to Land Nadir observations.
5
