# Peer review of "Title: Evaluation And Attribution Of OCO-2 XCO₂ Uncertainties"

_Atmospheric Measurement Techniques, 2016_

## Referee Comment (RC1) · Anonymous Referee #1 · 5 Oct 2016

**Review of Worden et al, "Evaluation and Attribution of OCO-2 $XCO_2$ Uncertainties"**

In this manuscript, the authors investigate whether the variation in OCO2 $XCO_2$ is consistent with the error statistics reported by the ACOS retrieval algorithm. They consider the variation of retrieved $XCO_2$ within a "small" area of ~100 x 10.5 km$^2$ (~14 sec of ground track), and check whether the statistics of that variation are consistent with (a) random error, (b) correlated random error, or (c) a slowly varying bias from non-$CO_2$ elements of the retrieval state vector. Their approach is systematic and well elucidated, and yields the not surprising conclusion that a slowly varying bias from interference terms is a key component of $XCO_2$ variation over small areas. The numbers they derive for the effective precision and accuracy of land and ocean soundings are reasonable.

I have one major comment, and several minor comments. If the authors can respond to these satisfactorily (especially the major comment), I would recommend publication of the manuscript in AMT.

**Major comment**

A key question the authors need to answer when considering the variation of retrieved $XCO_2$ over a small area is "how much of this is real?", since that variation must be "taken out" to quantify the contributing factors behind the remaining variation. To do this, the authors consider $XCO_2$ fields from CarbonTracker (CT), which is run at 1° x 1° over North America, which the authors call "high resolution". The N-S gradient of CT $XCO_2$ has an RMS of ~0.3 ppm/100 km, which the authors consider a plausible measure of flux- and transport-driven variability.

I disagree with their approach and conclusion for two reasons. First, the N-S gradient of a 1° x 1° model is not expected to mirror gradients in the real atmosphere, especially when the objective is to explain gradients seen by an instrument which takes soundings every ~2 km going from S to N. In that context, 1° x 1° is hardly "high resolution", despite what the authors claim, and CT $XCO_2$ is expected to be much smoother (and hence N-S gradients much smaller) compared to gradients at the scale observed by OCO-2. To illustrate my point, I've made plots analogous to Figure 1 of the manuscript, but taking $XCO_2$ from a NASA GMAO high resolution (~7m km globally) free running GEOS-5 $CO_2$ simulation available at https://gmao.gsfc.nasa.gov/global_mesoscale/7km-G5NR/data_access/. The modeled $XCO_2$ within a box enclosing the contiguous United States was sampled at 20:00 UTC to be close to the 13:30 local overpass time of OCO2 over the geographical center of the US. While the fluxes in

this model are not optimized, they are realistic. More importantly, the transport-induced variability over short spatial scales is expected to be more realistic than in CT. As can be seen in Figure R1 below, the N-S gradient in the model is highly dependent on the resolution, and is in general much higher than those evaluated from CT by the authors.

[Figure]

**Figure R1:** Histograms of north-south gradients of $XCO_2$ calculated from a ~7 x 7 km² model run at NASA GMAO. The model fields are available at 1/16 and 1/2 degree resolutions. In both cases, the fields were sampled at 20:00 UTC (corresponding to 13:30 local time in the center of the conterminous US), and the histograms represent all possible N-S gradients within a box (24°N to 50°N, 127°W to 64°W) covering the conterminous US. The modeled field at a higher resolution has steeper N-S gradients, much higher than the gradients from 1° x 1° CT.

Second, I would argue that for the authors' purpose what is important is not the N-S gradient but rather the variability of $XCO_2$ in the real atmosphere within a ~100 x 10.5 km² area. This is impossible to get from 1° x 1° CT fields, since all of that area is within a single grid cell. I evaluated that variability using the previously mentioned 0.5° and 0.0625° model fields in Figure R2. At 0.0625°, which is still coarser than the OCO2 footprint, a significant fraction of the small areas considered had variability larger than 0.4 ppm over land. Since any Eulerian model variability is limited by numerical diffusion, we can expect that the real atmosphere has even more variability at the ~2 km length scale commensurate with OCO2 pixels.

All this is to say that the variation of OCO2 $XCO_2$ seen by the authors within each small area could be entirely explained by variability in the real atmosphere, and perhaps the authors don't see that because they look at the N-S gradient (not the variability) of a fairly coarse resolution 1° x 1° model. Their assertion on page 3, lines 21-23 ("the expected variability in $XCO_2$ …

comparable or less than the calculated OCO-2 uncertainties") may not hold for the real atmosphere.

[Figure]

**Figure R2:** The standard deviation of modeled $XCO_2$ over a ~100 km N-S stretch for two model resolutions. The models were subsampled analogous to Figure R1 to adhere closely to what might be observed by OCO-2. As model resolution is increased, the modeled variability increases. It is expected that OCO-2, with a footprint of ~1.25 km x 2 km, will observe an even higher variability than a 7 km x 7 km model.

I would like the authors to respond to this argument, i.e., what would happen to their estimate of the different factors behind the variation of $XCO_2$ over small areas, if it turned out that their CT-derived estimate of the atmospheric variability was too low, and in fact the real atmospheric variability was high enough to explain all of the OCO-2 observed variability?

**Minor comments**

1. P4, L5: It is not correct to say that the OCO-2 instrument always observes the "glint spot" of specular reflection, since, as the very next sentence explains, there are both "glint" and "nadir" modes.
2. P4, L15: How do the authors know that the statistics of the target mode soundings are spurious? What makes them spurious?
3. P4, L24: The authors use bias-corrected $XCO_2$ for this exercise. In theory, bias correction should remove long-range correlations in the error in $XCO_2$ by reducing what the authors call interference error. However, if the bias correction parameters are not chosen correctly, the bias correction itself will introduce slowly varying biases. Can the authors verify that using non-bias corrected (or "raw") $XCO_2$ from ACOS leads to a larger estimate of the slowly varying error in H3?

4. P4, L25: As far as I know, the bias correction depends not just on TCCON $XCO_2$ but as well as on the so-called "southern hemisphere approximation" and a small area analysis where $XCO_2$ is assumed constant over < 100 km along track (see page 14 of http://disc.sci.gsfc.nasa.gov/OCO-2/documentation/oco-2-v7/OCO2_XCO2_Lite_Files_and_Bias_Correction.pdf).

5. P4, L28-29: There is emerging consensus in the OCO-2 flux inversion community that filtering by warn levels (WL) only lets in retrievals with significant bias from interference terms. Rather, filtering by **xco2_quality_flag**, which is WL < 15 plus some additional criteria on retrieved aerosol and $CO_2$ parameters, is a much better way of reducing the number of biased samples. Can the authors confirm this by showing that if they use soundings with **xco2_quality_flag** = 0 they get a smaller contribution from the slowly varying bias of H3?

6. P4, L29: The highest WL is 19, not 20.

7. P5, L13: Should be "$XCO_2$" instead of "$_{XCO2}$"

8. P5, L21: The CT-based variability in the N-S gradient of $XCO_2$ was estimated only over North America, yet it seems to have been used everywhere between 30°S and 30°N. How valid is this assumption?

9. P7, L11-13: In the statement of hypotheses, I think the authors mean "variations in $XCO_2$" and not "uncertainties". If I understand correctly, the entire point of the manuscript is to see whether variations in $XCO_2$ within a small area are consistent with $XCO_2$ errors being primarily from random noise, correlated noise, or a slowly varying bias. So the choice of words in L11-13 is important, and I'd like the authors to either confirm or refute my understanding that "uncertainties" should be replaced by "variations in $XCO_2$".

10. P9, L18-20: Can the onset of this strong inverse relationship between calculated and actual uncertainty below a certain threshold be used to filter out seemingly low noise (high SNR) soundings over the tropical oceans that might be biased?

11. P10, L1: I think the authors mean "Figure 2" (or 3, or 4) instead of "Figure 1".

12. P11, L3: Why the lag of 0.3 sec? Is it because OCO-2 cross-track "strips" are spaced 0.3 sec apart along track? If so, that should be mentioned.

13. P12, L18-20: Recent results shown at OCO-2 science team meetings and telecons suggest that over small areas, surface elevation has a strong impact on retrieved $XCO_2$. Is this included in **GK**$_y$, i.e., is surface elevation in the vector **y**?

14. P13, L1: Each of the distributions (Gaussian, Lorentz, Laplace) considered by the authors has a physical basis, i.e., there are reasons why a quantity might follow one of the three distributions. E.g., if two independent variables each follow an exponential distribution, then their difference follows a Laplace distribution. Can the authors speculate why the slopes in Figure 8 might behave like such a quantity?

---

## Referee Comment (RC2) · Anonymous Referee #2 · 14 Oct 2016

see supplement

Please also note the supplement to this comment:
http://www.atmos-meas-tech-discuss.net/amt-2016-175/amt-2016-175-RC2-supplement.pdf

---

## Author Comment (AC1) · 14 Nov 2016

We would like to thank the reviewers for their comments, especially reviewer 1 for their analysis on modeled XCO2 variations at very fine length scales.

We have repeated reviewer 1's analysis and while we get different numbers for the distribution of slopes (we get ∼1 ppm) versus their ∼ 2ppm over land), we do get similar distributions for the STD within a small area.

As Reviewer 1 indicates this means that one of our primary assumptions, that XCO2 varies less than the uncertainties no longer applies.

On the one hand this means that the observed XCO2 variability is likely due to real XCO2 variations. This is a good result for OCO-2! Furthermore, our result about the

measurement error (from noise and natural variability) is still valid.

On the other hand, it is much more challenging to bound the role of interference error within a small neighborhood.

We are therefore looking at two approaches 1) Evaluating the correlation length scales of potential interferences by using the OCO-2 retrieved and a priori and asking at what point they are de-correlated from XCO2 variations and 2) Asking if we can provide an upper bound on the role of interferences within a small neighborhood.

We therefore request more time (until after the holidays) to re-evaluate these uncertainties and furthermore request the same reviewers given that they are already vested in the analysis.
* * *

---

## Author Comment (AC2) · 16 Apr 2017

Response to Reviewer 1

We would like to thank the reviewer for their analysis of the GMAO data which we believe greatly improved the analysis discussed in our paper. We apologize for the delay in re-submitting the paper but as discussed next it took some time to obtain the needed GMAO fields, replicate as best as possible the reviewers' results, and propagate these results to our analysis.

In order to address the reviewers concern we downloaded the high resolution GMAO $CO_2$ data for the spatio-temporal matchups corresponding to OCO-2 measurements used in our analysis. Note that the OCO-2/GMAO data is for 2005-2007, which does not match the OCO-2 observation timeframe, but we would expect the size of the variations to be adequately captured by comparing to the same locations and times from a different year. We repeated as best as possible both the analysis of the reviewer and examined the variability corresponding of the model corresponding to the data. The variations we found, even when matching what the reviewer did, were less than the variability the reviewer found. We show our findings at the end of this response. Our conclusions are therefore effectively unchanged as on average the "true" variability represented by the modeled $XCO_2$ is still much less than that observed by OCO-2.

To address this concern in the paper we have moved part of the introduction, where we had previously discussed the role of natural variability, into the analysis section and added the hypothesis "*H1: Observed variability is due to natural $XCO_2$ variability*". We have also added an Appendix where we looked at the expected distributions of observed $XCO_2$ when accounting for natural variability, calculated noise, and calculated interference error and compared them to what is observed before and after the bias correction.

As mentioned above, a critical caveat to our update is that the variability we find from the modeled fields is less that that found by the reviewer (see Figure below). We have triple checked our book-keeping so we would like to ask the reviewer to check theirs to ensure our results make sense.

Minor Comments

1. P4, L5: It is not correct to say that the OCO-2 instrument always observes the "glint spot" of specular reflection, since, as the very next sentence explains, there are both "glint" and "nadir" modes.

   Response:  Fixed (grammar error)

2. P4, L15: How do the authors know that the statistics of the target mode soundings are spurious? What makes them spurious?

   Response: Brought cited papers up to front of reference and changed spurious to "outside of the expected range"

3. P4, L24: The authors use bias-corrected $XCO_2$ for this exercise. In theory, bias correction should remove long-range correlations in the error in $XCO_2$ by reducing what the authors call interference error. However, if the bias correction parameters are not chosen correctly, the bias correction itself will introduce slowly varying biases. Can the authors verify that using non-bias corrected (or "raw") $XCO_2$ from ACOS leads to a larger estimate of the slowly varying error in H3?

   Response: We have added discussion on how the bias correction affects the observed variability in the appendix and as the reviewer suggests it does improve the comparison. Our analysis shows that the bias correction REDUCES the slowly varying gradients as one might expect if the bias correction is correcting errors related to aerosols and because we expect the gradient is caused by an issue related to the aerosols.

   More discussion on the bias correction is in the Wunch et al. papers (cited)

4. P4, L25: As far as I know, the bias correction depends not just on TCCON $XCO_2$ but as well as on the so-called "southern hemisphere approximation" and a small area analysis where $XCO_2$ is assumed constant over < 100 km along track (see page 14 of http://disc.sci.gsfc.nasa.gov/OCO-2/documentation/oco-2-v7/OCO2_XCO2_Lite_Files_and_Bias_Correction.pdf ).

   Response: Fixed language

5. P4, L28-29: There is emerging consensus in the OCO-2 flux inversion community that filtering by warn levels (WL) only lets in retrievals with significant bias from interference terms. Rather, filtering by **x co2_quality_flag** , which is WL < 15 plus some additional criteria on retrieved aerosol and $CO_2$ parameters, is a much better way of reducing the number of biased samples. Can the authors confirm this by showing that if they use soundings with **xco2_quality_flag** = 0 they get a smaller contribution from the slowly varying bias of H3?

   Response:  I would prefer to keep more detailed, iterative studies like this off-line and instead use the analysis shown in this paper as well as those from Wunch et al. [2016] and Connor et al. [2017] as descriptions of methods and initial results used to test these refined hypothesis about the uncertainties.

6. P4, L29: The highest WL is 19, not 20.   (Fixed)

7. P5, L13: Should be "XCO$_2$" instead of "$X_{CO2}$"   (Fixed)

8. P5, L21: The CT-based variability in the N-S gradient of XCO$_2$ was estimated only over
   North America, yet it seems to have been used everywhere between 30°S and 30°N.
   How   valid is this assumption?   (Addressed with major comment)

9. P7, L11-13: In the statement of hypotheses, I think the authors mean "variations in XCO$_2$"
   and not "uncertainties". If I understand correctly, the entire point of the manuscript is to
   see whether variations in XCO$_2$ within a small area are consistent with XCO$_2$ errors
   being primarily from random noise, correlated noise, or a slowly varying bias. So the
   choice of words in L11-13 is important, and I'd like the authors to either confirm or
   refute my understanding that "uncertainties" should be replaced by "variations in XCO$_2$"
   .  Response: Changed

10. P9, L18-20: Can the onset of this strong inverse relationship between calculated and
    actual uncertainty below a certain threshold be used to filter out seemingly low noise
    (high SNR) soundings over the tropical oceans that might be biased?

    Response: possibly and we have communicated this issue to the OCO-2 team (several of
    who are co-authors).

11. P10, L1: I think the authors mean "Figure 2" (or 3, or 4) instead of "Figure 1".   (Fixed)

12. P11, L3: Why the lag of 0.3 sec? Is it because OCO-2 cross-track "strips" are spaced 0.3
    sec   apart along track? If so, that should be mentioned.   (Fixed)

13. P12, L18-20: Recent results shown at OCO-2 science team meetings and telecons suggest
    that over small areas, surface elevation has a strong impact on retrieved XCO$_2$. Is this
    included in $\mathbf{GK}_y$, i.e., is surface elevation in the vector $\mathbf{y}$ ?

    Response:. Isnt this the same as an error in surface pressure? If so my understanding is
    that this error is included.

14. P13, L1: Each of the distributions (Gaussian, Lorentz, Laplace) considered by the authors

has a physical basis, i.e., there are reasons why a quantity might follow one of the three distributions. E.g., if two independent variables each follow an exponential distribution, then their difference follows a Laplace distribution. Can the authors speculate why the slopes in Figure 8 might behave like such a quantity?

Response: We (the authors) discussed why one shape or another had a better fit but could not come up with any reasonable explanation / hypotheses. For this reason we do not speculate in this paper the reason for the shape of the distribution. However, an update to the Connor et al. (2016) analysis which uses the same small neighborhoods we use, along with the observed distribution of these shapes could shed light into the primary sources of uncertainty in the $XCO_2$ data that are not currently accounted for by the uncertainty calculations. We have added a statement to that effect in the conclusion.

Response to Reviewer 2

**Comment**: Most of the issues concerning this paper have already been flagged by referee#1 and I will not repeat them here. However an additional point that I feel needs more elaboration is the difference in land and ocean results. When testing the first hypothesis, we see a 0.4 ppm bias shift between the observed land data and what could have been expected from the calculated measurement uncertainties. However if we look at the ocean data we see no bias in the >0.4 ppm calculated uncertainty bins. Are the potential components that yield the land bias [synoptic variation, non-linearities in the retrieval, etc] truly all absent over the ocean? If not, the calculated error components over the ocean might very well be overestimated. In the summary this is again touched upon. Calculated ocean errors are simply deemed correct, while land (including glint-land) are deemed to be underestimated. The authors need thus to explain why the potential sources of error play no (or insignificant) role in the glint ocean retrievals.

Response: Mechanistically explaining why these differences exist is well beyond the scope of this study as it requires systematically updated a sophisticated radiative transfer model in order to test each effect. Rather we document the differences here which can then be used to test the uncertainties in a future study. We add a statement in the abstract and the summary that these ocean/land differences lead us to suspect that surface pressure and albedo are the likely issues affecting the accuracy of OCO-2 data because we would expect surface pressure and albedo to vary more strongly over land than ocean. We have added a sentence in the abstract and the following statement in the summary to address these issues:

This analysis sheds further light on the sources of uncertainty of the observed $XCO_2$ data. For example, the $XCO_2$ gradient variability in the small neighborhoods over the ocean as compared to the land suggests that the largest uncertainty in OCO-2 $XCO_2$ data is related to surface properties such as surface pressure or albedo because we expect larger variations of these geophysical parameters over land. The observed gradients could also be related to the variation in solar zenith angle as OCO-2 data takes observations because the effect is manifested as a slowly varying quantity in addition to increased random variability. The observed distribution of these $XCO_2$ gradients over the whole globe, which has a Laplace distribution, is also a potential clue as any bottom-up or future analysis that attempts to model the $XCO_2$ uncertainties should also replicate this distribution. A future study in which the calculated uncertainties for OCO-2 discussed in Connor *et al.* (2016) repeats the steps shown in this paper in conjunction with the OCO-2 / TCCON data will hopefully reveal and characterize the likely sources of these uncertainties.

Minor comments

P1L28: ...in reasons that 'are' not well understood (fixed)

P4L3-4: remove 'with' and 'observes'... (fixed)

P12L25: the calculated random noise or 'actual' noise? (fixed)

**Expanded Discussion on XCO₂ variability in response to Reviewer 1 Comments**

Model analysis from GMAO 7 km model fields

Again, we thank reviewer 1 for pointing us to this model output which is much more relevant to our analysis than the 1 x 1 degree CarbonTracker output. We looked at the 7 km data for July 7, 2006 at UTC 20:00 from 24-50N and 127-64W (file ftp://ftp.nccs.nasa.gov/Ganymed/7km/c1440_NR/DATA/0.0625_deg/inst/inst30mn_3d_CO2_Nv/Y2006/M07/c1440_NR.inst30mn_3d_CO2_Nv.20060707_2000z.nc4). The data was classified as land or water using a land surface map from the UW/CIMS infrared emissivity database (Vidot and Borbas, 2014). XCO2 was calculated based on pressure weighting with pressure from the corresponding pressure file (c1440_NR.inst30mn_3d_DELP_Nv.20060707_2000z.nc4). Every 0.2 degrees latitude and longitude, a pair of points spaced by 15 points (or 105 km) north/south are selected. A histogram of the differences of XCO2 between these points was plotted, selecting either ocean or land subsets. The same analysis was also performed for the observations used in the paper, which span September, 2014 to May, 2015 and do not include summer, which has the most variability.

[Figure]

Figure 1. 100 km N/S $XCO_2$ differences for land (red) and ocean (blue) from NASA GMAO 7 km run for XCO2 for Summer 2006 (left) and locations/times matching OCO-2 observations (offset by year) (right)

The $XCO_2$ north/south 100 km differences found by the reviewer is 0.8 ppm (land) and 0.4 ppm (ocean) for the 0.5 degree resolution. For the 7 km resolution, the reviewer found 2.2 ppm (land) and 0.9 ppm (ocean). Our results for the 7 km resolution GMAO model agree with the reviewer's result for the 0.5 degree resolution but show much smaller distribution of gradients than the reviewer's results for the 7 km resolution. Our results for the 7 km resolution GMAO model have about, on average, twice the gradient as seen in the 1x1 degree CarbonTracker model shown in Fig. 1 of the discussion paper and Figure 2 below. However, when matched to the observation locations/times used in this paper, gradients of 0.2 ppm to 0.4 ppm are seen, which is smaller than the variability seen in the OCO-2 data shown in Fig. 7 of the discussion paper (now Figure 8). Therefore the conclusion that the larger variability and gradients seen in the OCO-2 data do not result from natural variability is not changed by the additional study of the high resolution model fields.

[Figure]

Figure 1: Distribution of latitudinal $XCO_2$ gradients as calculated by the high resolution, "Real Time", Carbon Tracker model for November 2015 (left panel) and July 2015 (right panel) over North America and the nearby oceans. The latitude grid is 1 degree or ~110 km. The gradients are re-scaled to 100 km for comparison to the $XCO_2$ gradients discussed in this paper.

Response to Editor (Ilse Aben) Comments

**Comment**: It seems that sometimes slightly different terminology is used for the same thing. This unnecessarily complicates the reading and understanding of the paper. It would be very helpful to stick to the same terminology throughout the paper. (an example : p.8-9 calculated measurement noise, calculated measurement error, measurement uncertainty due to noise, are these all different things or are they indicating the same thing ? If they mean the same thing please use one term)

**Response**: I have replaced all instances of "measurement error" with "measurement uncertainty" and where appropriate added a caveat, i.e. measurement uncertainty due to noise (as opposed to interferences).

**INTRODUCTION**

**Comment**:- The analysis with Ctracker is limited to the US, which means roughly speaking latitudes higher than 30 N. Whereas the OCO-2 data analyses that follows focuses mostly on neighborhoods between 30S-30N. To what extent are the variabilities as obtained for N-America then useful to compare with ?

**Response**: We have removed the comparison with Ctracker and replaced with GMAO

**Comment**: p.3,l.16 'while in-situ ...', are these column integrated variabilities or are these in-situ measurements at a certain height or .... ?   - p.3, l.16 'while in situ and model data ..' what model data do you refer to here ?

**Response**: This should be fixed in the subsequent version as we now exclusively compare to the GMAO model fields

**Comment: OVERVIEW OF THE OCO-2 DATA**   - p. 5 I am bit lost now. In the refered document a description is given on the Bias correction for OCO data. Here also corrections are based on the small areas and variability seen within these small areas, and a correction based on main parameters influencing that. Has such a correction already been applied to the data here ? If so, how does that affect the neighborhood analises here ?   - for which period OCO-2 data is analised ? I don't think it is mentioned anywhere. I think it should be mentioned quite early in the paper.

**Response**: We added language in this section stating that the analysis is bias corrected data and how it is bias corrected. We also show in the Appendix how the bias correction improves the comparison between expected and actual variability.

**EVALUATION OF UNCERTAINTIES**

**Comment**: L17-19, p.5 the data that is used in a neighborhood is presumably taken from one orbit and are thus very close in time ? So never data from different moments in time that happen to fall in the same 100 x10.5 km area are compared as part of one neighborhood ?

**Response**: Correct. We have added "taken consecutively" when describing the data in a neighborhood for clarification.

**Comment**:L19-22 p. 5 please provide map to show the locations of these neighborhoods. If not in the paper than at least in the response such that it is visible to interested readers.

**Response**: Maps of the neighborhoods for each observing mode are provided below. As stated in the text most neighborhoods where there are at least 50 consecutive data points that also passed the quality flags are in the sub-tropics.

**Comment**: I think the basic info to verify the average 190 observations are not in the manuscript. Please provide, and briefly explain why indeed you have ~190 observations per neighborhood on average to work with. Is there also a way that we can understand why you get roughly 39000 neighborhoods to work with ?

**Response**: Within about 100km (the size of the neighborhood), OCO-2 takes approximately 190 observations. We added language to clarify

**Comment**: L. 27 mean CO2 column à mean CO2 column in a neighborhood ?  (yes, fixed)

**Comment**: P. 7 is 'small area' the same as 'small neighborhood' ? if so please use one   term throughout the paper.  (fixed)

**Minor textual**: (all fixed)

[Figure]

[Figure]

[Figure]

Land Glint (WL <= 10, Bias Corrected)